# Antinociceptive Actions of Botulinum Toxin A1 on Immunogenic Hypersensitivity in Temporomandibular Joint of Rats

**DOI:** 10.3390/toxins14030161

**Published:** 2022-02-23

**Authors:** Victor Ricardo Manuel Muñoz-Lora, Ana Dugonjić Okroša, Ivica Matak, Altair Antoninha Del Bel Cury, Mikhail Kalinichev, Zdravko Lacković

**Affiliations:** 1Laboratory of Molecular Neuropharmacology, Department of Pharmacology, School of Medicine, University of Zagreb, 10000 Zagreb, Croatia; victor_9874@hotmail.com (V.R.M.M.-L.); ivica.matak@mef.hr (I.M.); 2Department of Prosthodontics and Periodontology, Piracicaba Dental School, University of Campinas (UNICAMP), Piracicaba 13414-903, Brazil; altair@unicamp.br; 3Dental Research Division, School of Dentistry, Ibirapuera University, São Paulo 04661-100, Brazil; 4Department of Pharmacology, Faculty of Pharmacy and Biochemistry, University of Zagreb, 10000 Zagreb, Croatia; anadugonjic2008@gmail.com; 5Ipsen Innovation, 91940 Les Ulis, France; mikhail.kalinichev@icloud.com

**Keywords:** botulinum neurotoxin type A1, nociception, microglia, astrocytes, central nervous system, analgesia

## Abstract

Botulinum neurotoxin type A1 (BoNT-A) reduces the peripheral peptide and cytokine upregulation in rats with antigen-evoked persistent immunogenic hypersensitivity (PIH) of the temporomandibular joint (TMJ). Herein, we examined the effects of two preparations of BoNT-A, abobotulinumtoxinA (aboBoNT-A; Dysport) and onabotulinumtoxinA (onaBoNT-A; Botox), on spontaneous and evoked nociceptive behaviors, as well as on central neuronal and astroglial activation. The antigen-evoked PIH was induced in rats via repeated systemic and unilateral intra-articular (i.a.) injections of methylated bovine serum albumin (mBSA). Rats were subsequently injected with unilateral i.a. aboBoNT-A (14 U/kg), onaBoNT-A (7 U/kg), or the vehicle (saline). After i.a. treatments, spontaneous and mechanically evoked nocifensive behaviors were assessed before and after the low-dose i.a. formalin (0.5%) challenge. The central effects of BoNT-A were assessed by an immunohistochemical analysis of cleaved synaptosomal-associated protein 25 (cSNAP-25) presence, c-Fos, GFAP, and CGRP expression in the trigeminal nucleus caudalis (TNC). Both BoNT-A preparations similarly reduced the formalin-induced spontaneous pain-related behaviors and mechanical allodynia of the hypernociceptive rats. Likewise, their effects were associated with the central occurrence of cSNAP-25 and reduction of c-Fos and GFAP upregulation in the TNC. BoNT-A antinociceptive activity on the PIH is associated with the toxin axonal transport to trigeminal sensory areas and reduction of neuronal and glial activation in central nociceptive regions.

## 1. Introduction

Temporomandibular joint (TMJ) rheumatoid arthritis (RA) is a chronic inflammatory condition involving a range of local and systemic proinflammatory mediators. TMJ-RA produces structural and functional deterioration of the affected TMJ and associated tissues [1]. TMJ-related symptoms are present in almost 65% of RA patients and are typically bilateral [2,3]. Along with the impairment of normal functions of the jaw joint, the TMJ-RA is often associated with periods of alternating nonpainful and painful phases [1]. Active painful phases are usually provoked by an otherwise nonpainful stimulus (e.g., mandibular functional movements: speaking, eating, etc.), with a negative impact on the patient’s quality of life [4,5]. Notwithstanding, treatments with nonsteroidal anti-inflammatory drugs (NSAIDs) and local anesthetic into the TMJ compartment have proven to relieve RA pain just temporarily [6].

Animal models of RA-related pain, including TMJ-RA, have been proposed with the goal to improve the understanding of its etiology and develop new treatment options [7,8,9]. When used as an antigen, systemic and unilateral intra-TMJ administrations of methylated bovine serum albumin (mBSA) in rats result in the development of persistent immunogenic hypersensitivity (PIH) and inflammatory changes of the exposed TMJ (Figure 1) [10,11]. Interestingly, even though this model resembles human TMJ-RA, it has a monoarthritic (single joint) presentation, since it is induced locally inside the mBSA-stimulated joint [8,10,11]. The hypersensitivity appears as an exaggerated pain-related response (facial grooming and head flinching) to a normally nonpainful chemical stimulation of the TMJ, such as a low-dose (0.5%) formalin challenge [8,12]. This arthritic model produces a delayed-type hypersensitivity (viz., type IV hypersensitivity) and is characterized by central sensitization due to immune and neuronal cells activation, leading to the local release of a variety pronociceptive factors [10,11,13,14]. This fact increases the validity of the model to study the progression of pain on chronic inflammatory conditions and to test promising treatment options.

In animals exposed to PIH and challenged with a low-dose formalin in the TMJ, we previously demonstrated that a treatment with an intra-TMJ injection of botulinum neurotoxin type A1 (BoNT-A), a potent neurotoxin naturally produced by *Clostridium botulinum*, reduced spontaneous behavioral facial grooming and head flinching responses 24h and 14 days after its administration [8]. Additionally, BoNT-A reduced the upregulated levels of substance P and calcitonin gene-related peptide (CGRP) in the trigeminal ganglion and interleukin 1 beta (IL-1β) in the TMJ of exposed rats [8]. Despite the apparent association of BoNT-A antinociceptive action with the mentioned peripheral effects, numerous studies in different pain models have suggested a necessity for an axonal transport and a direct central analgesic action of the toxin [15,16]. Therefore, we examined the occurrence of cleaved SNAP-25 (cSNAP-25), along with nociceptive neuronal and astroglial activation, on the trigeminal brainstem sensory areas of rats induced to PIH and treated with BoNT-A. In addition, spontaneous and evoked nociceptive behaviors were also assessed, and the efficacy of two different commercially available pharmaceutical preparations of BoNT-A: abobotulinumtoxinA (aboBoNT-A) and onabotulinumtoxinA (onaBoNT-A) administered at equiefficatious doses were compared.

## 2. Results

### 2.1. BoNT-A Effects on Rat Grimace Scale (RGS) and Spontaneous Nocifensive Responses (Flinching, Sctratching)

Previously in the PIH model, the number of spontaneous motor responses (flinches and scratching) after the formalin stimulation was shown to be reduced by onaBoNT-A [8]. Herein, we extended the measurement of spontaneous nociception by examining facial grimacing related to pain (RGS) and examined if the spontaneous nociceptive responses are present both before and after TMJ stimulation with low-dose formalin 0.5%). In the RGS test, we found a lack of painful facial expression (score = 0) and a low number of behavioral nocifensive responses in all groups at the pre-formalin assessments. Conversely, the post-formalin assessments showed higher values of RGS (Figure 2A,B) and a significant increase in the nocifensive responses (Figure 2C) on induced animals compared to noninduced.

Compared to induced animals treated with saline, both BoNT-A groups exhibited a significant reduction of the nocifensive responses triggered by formalin (Figure 2C). Furthermore, both BoNT-A treatments resulted in a similar reduction (*p* < 0.01) of the RGS score after formalin injection (Figure 2B).

### 2.2. Effects of BoNT-A on Mechanically-Evoked Responses at the TMJ Area

By employing the von Frey filaments, we examined the possible occurrence of facial mechanical allodynia over the skin covering the stimulated TMJ area both before and after the stimulation with 0.5% formalin. All animals showed a lack of allodynic responses at the pre-formalin evaluation. Following the formalin challenge, PIH-induced animals developed bilateral mechanical allodynia over the area of TMJ, measured with von Frey filaments (Figure 3A,B). The after-formalin assessments showed that BoNT-A (aboBoNT-A and onaBoNT-A) reduced the mechanically evoked TMJ bilateral allodynia of the hypersensitive rats (Figure 2A,B).

### 2.3. Effect of BoNT-A on Neuronal and Astrocyte Activation in the TNC

We further examined the possible BoNT-A effects on neuronal and astrocyte activation associated with nociception in the trigeminal nucleus caudalis (TNC), the first-order pain processing sensory nucleus of the cranial area. We assessed the post-formalin neuronal activation of second-order nociceptive neurons in the TNC by employing c-Fos immunohistochemistry (Figure 3A,B) and the counting of c-Fosexpressing neuronal profiles (Figure 2C). After a normally nonpainful formalin injection, induced animals presented high levels of c-Fos-positive nuclei in both ipsilateral and contralateral TNC. Unilateral applications of BoNT-A (aboBoNT-A or onaBoNT-A) significantly decreased post-formalin c-Fos expression in both ipsilateral and contralateral TNC (Figure 4B,C).

Analysis of TNC showed an ipsilateral increase in astrocytic marker glial fibrillary acidic protein (GFAP) immunoreactivity on induced rats compared to noninduced animals (Figure 5A). BoNT-A reduced the GFAP-immunoreactive area 14 days after treatment (Figure 5B,C). The levels of calcitonin gene-related peptide (CGRP) expression were not increased in the hypernociceptive rats or altered by BoNT-A (aboBoNT-A or onaBoNT-A) treatments (Figure 6).

### 2.4. Immunohistochemical Localization of cSNAP-25 in the Brain

Previously, we discovered that the BoNT/A antinociceptive action in the craniofacial area is associated with the occurrence of cleaved SNAP-25 (cSNAP-25) in the TNC following axonal transport [17]. Thus, we further examined if the antinociceptive actions of aboBoNT/A and onaBoNT/A are similarly associated with the direct toxin activity in the TNC. Fourteen days after BoNT-A (aboBoNT-A or onaBoNT-A) i.a. injections into the left TMJ, BoNT-A cSNAP-25-positive nerve fibers were found in the ipsilateral trigeminal sensory regions (TNC). Similar cSNAP-25 staining was observed in both aboBoNT-A and onaBoNT-A-treated animals. cSNAP-25 on the contralateral (nontreated) side was not observed (Figure 7).

## 3. Discussion

In rats induced to PIH of the TMJ, we demonstrated that the analgesic activity of BoNT-A, expressed as a reduction of spontaneous and evoked nocifensive behaviors after a low-dose formalin stimulation, is associated with a decrease of neuronal and astroglial activity in the TNC. Moreover, in the same arthritis model, we previously demonstrated that BoNT-A reduced the nocifensive behaviors, as well as the expression of proinflammatory mediators and peptides in the TMJ and trigeminal ganglion [8], in accordance with other studies [18,19,20]. Notwithstanding, a causal role between BoNT-A antinociceptive activity and its peripheral effects has not been established. Furthermore, it was suggested that the BoNT-A analgesic effect necessarily involves an axonal transport of the toxin to central nervous system (CNS), where it may interact with neuronal and/or non-neuronal cells [15,21].

### 3.1. Peripherally Injected BoNT-A Reduces Spontaneous and Mechanically Evoked Nocifensive Behaviors in Rats with PIH

Despite the existence of numerous tests for animals’ behavior assessment [22], spontaneous behavioral assessments may be more relevant than stimulus-evoked assessments for future clinical translations, increasing the validity of the model [7]. However, the orofacial region remains an underrepresented area in spontaneous pain behavioral assessments. Persistent scratching and rubbing of the facial region after chemical or inflammatory stimulation (e.g., the orofacial formalin test) is used as a sign of pain in freely moving rodents [23,24]. For this reason, we also assessed the spontaneous nociceptive behaviors by employing RGS. Since pain is a well-characterized emotion based on the facial action coding system [25], the RGS is capable of detecting spontaneous nociception [22] and measuring the analgesic efficacy of drugs [26].

The arthritic rats showed no visible signs of evoked or spontaneous pain prior to formalin TMJ stimulation. However, low-dose formalin (0.5%), known not to induce pain-related behaviors in normal animals [27], induced a significant increase of facial grimacing, head flinching, facial rubbing, and bilateral mechanical allodynia (Figure 2 and Figure 3), in accordance with previous studies [8,28]. This suggests that the mBSA-induced monoarthritis is nonpainful per se; however, it induces a profound chemical hypersensitivity to normally nonpainful concentrations of inflammatory substances such as formalin [12]. This exacerbation of nociceptive responses by normally nonpainful chemical stimuli suggests a quick triggering of central sensitization processes [10].

BoNT-A decreased the flinching and rubbing nocifensive responses (Figure 2C) and reduced the facial nociceptive grimacing of rats exposed to PIH in the TMJ. Importantly, the RGS was quantified at three different periods after formalin stimulation in the TMJ. Shortly after the formalin injection (0–5 min), BoNT-A was not effective in RGS reduction, similar to the observed lack of BoNT-A action on immediate formalin-evoked pain sensation (phase I of formalin test) [17,19]. However, at later time points (9–14 and 18–23 min), both BoNT-A groups significantly reduced the spontaneous pain grimacing score (Figure 2B), demonstrating BoNT-A action in the tonic pain phase when central sensitization takes place.

In addition, we assessed the mechanical allodynia by von Frey monofilaments prior and after 0.5% formalin stimulation of the exposed TMJ. Allodynia over the TMJ area in PIH-induced rats was observed only after the 0.5% formalin challenge at both sides (ipsi- and contralateral TMJ). This bilateral allodynia was reduced by a unilateral BoNT-A treatment resembling the toxin’s bilateral action on experimental trigeminal neuropathic pain [29] and polyneuropathic pain [16].

### 3.2. Effects of BoNT-A on Nociceptive Neuronal and Glial Activation

Apart from BoNT-A behavioral antinociceptive activity, we examined its action at the level of central sensory nociceptive nuclei by exploiting c-Fos activation as a reliable marker of neuronal activity [23,30]. Interestingly, we found an occurrence of c-Fos activated neurons in the ipsilateral and contralateral TNC after 0.5% formalin stimulation of the TMJ area (Figure 4), which was not observed in the orofacial or hind paw formalin tests [31,32,33]. The unilateral stimulation of mouse masseter with complete Freund’s adjuvant (CFA) similarly induces bilateral expression of c-Fos in the TNC [23]. This suggests that pain in the TMJ region and related muscles may be associated with the bilateral activation of trigeminal nociception, similar to referred pain in humans [34,35]. All these results indicate that the pain triggered by formalin activates contralateral trigeminal nociceptive nuclei, explaining the occurrence of bilateral mechanical allodynia (Figure 3A,B). Importantly, both BoNT-A preparations (aboBoNT-A and onaBoNT-A) injected into the ipsilateral TMJ reduced both bilateral neuronal activation and the bilateral allodynia in the TMJ area.

In addition, we examined the possible changes in the activation of glial cells in the CNS, since they are key to initiate and maintain central sensitization after peripheral tissue/nerve injury [10,36,37]. Astrocytes represent the majority of glial cells and play a significant role in the persistence of pain [38]. Both BoNT-A preparations reduced GFAP (astrocyte marker) immunoreactivity measured by surface area immunoreactivity and grey intensity in the TNC of arthritic rats (Figure 5). An in vitro study showed a slight or no direct effect of BoNT-A on astrocytes [39]. Another in vivo study detected the presence of cSNAP-25 in spinal cord astrocytes of a mouse with induced neuropathy and treated in the hind paw with high-dose (15 pg/mouse) BoNT-A devoid of complexing proteins [40]. The reduced astroglial expression supports the hypothesis that BoNT-A may have a modulatory action on glial cells and their role in central nociceptive sensitization [20]. However, BoNT-A effects on glial cells (microglia and astrocytes) should be more carefully explored under pharmacologically relevant moderate doses.

Finally, we assessed possible changes in the peptidergic content in the central afferent terminals by examining the expression of CGRP in the TNC, which was previously found to be increased in the more acute CFA-evoked unilateral monoarthritis of the TMJ [41]. Herein, we did not observe changes in the CGRP expression in the TNC (Figure 6).

### 3.3. Localization of cSNAP-25 in TNC

Formerly, it was assumed that BoNT-A actions resided peripherally, and its antinociceptive activity was linked to a local reduction of the synaptic transmitter/proinflammatory peptide release [18,42]. However, its analgesic effects were shown to be mediated by axonal transport and associated with toxin’s enzymatic activity in the CNS [15,17,32]. Previously, we discovered that inhibition of the axonal transport within the peripheral nerve or in the sensory ganglion prevent the antinociceptive effect of peripherally injected BoNT-A, while the toxin injections in the peripheral nerve or in the trigeminal ganglion at doses lower than the ones effective in the periphery produce an antinociceptive effect [15,17,29]. Herein, we confirmed the occurrence of BoNT-A (aboBoNT-A and onaBoNT-A) in the brainstem after i.a. injections aimed at the TMJ (Figure 7). cSNAP-25 appeared as fiber-like processes in the ipsilateral TNC, which is the primary pain-processing area for the orofacial nociceptive input [43]. In accordance with previous studies, we observed no cSNAP-25 in the contralateral TNC, in line with our observation that the toxin is axonally transported primarily via sensory neurons ipsilateral to the BoNT-A injection site and their central projections [32].

A possible transcytosis of the toxin within the central neurons has also been proposed [40,44]. However, this was not corroborated in previous studies employing low to moderate non-systemic onaBoNT-A doses (5-15 U/kg), which suggests that BoNT-A enzymatic activity resides in central afferent terminals of transient receptor potential vanilloid 1-expressing neurons [32]. A lack of immunohistochemical evidence for contralateral occurrence of the toxin’s enzymatic activity might suggest an unknown indirect analgesic effect of the toxin on the contralateral side. The exact mechanism of the BoNT-A action on contralateral pain processing remains unknown.

### 3.4. Preclinical Comparison of the Analgesic Efficacy of Different BoNT-A Pharmaceutical Preparations

Currently, there is an intense debate regarding the differences in efficacy of various BoNT-A formulations [45,46,47]. Although both aboBoNT-A and onaBoNT-A are produced from identical Hall A strain synthesizing the A1 toxin subtype; they differ in the composition of complexing auxiliary proteins (onaBoNT-A is composed of the entire 900-kDa complex, while aboBoNT-A has a variable composition of complexing proteins per neurotoxin molecule), as well as the relative potency of the different products [46]. Despite the possibility that these differences also depend on the amount of active toxins in individual preparations [47], they have rarely been studied in both animals and humans. To compare the effect of aboBoNT-A vs. onaBoNT-A, we employed a 2:1 dose conversion ratio according to Scaglione et al. [45]. Following a study from Field et al., who calculated the picogram content of each international unit (U) (5.38 pg of 150-kDa neurotoxin protein for aboBoNT-A, while 1 U of onaBoNT-A contains 9 pg of 150-kDa protein) [47], approx. 400-g rats were herein injected with similar 150-kDa BoNT-A amounts (30.1 pg/animal of aboBoNT-A and 25.2 pg/animal of onaBoNT-A). Despite some possible differences in the total amount of the neurotoxin injected, we found no significant differences in the antinociceptive activity of any investigated outcome between the two BoNT-A preparations (Figure 2, Figure 3, Figure 4, Figure 5 and Figure 6). This suggests that both products may have similar analgesic effects [48,49].

### 3.5. Limitations

As a limitation of the study, we must consider that this experiment is restricted to the trigeminal region, so differences in the mechanism of action of BoNT-A applied to other extracranial joints (e.g., knee and ankle) and actions at the spinal levels cannot be excluded. Additionally, during the period after BoNT-A treatment, the aboBoNT-A- or onaBoNT-A-treated rats did not exhibit a significantly slower weight gain compared to the saline-treated rat groups (*p* = 0.070, one-way ANOVA, results not shown). This excludes possible systemic effects or a local muscular BoNT-A action (e.g., mastication and feeding) interfering with the animal weight gain.

## 4. Materials and Methods

Animal procedures were approved by the Ethics Committee of the University of Zagreb School of Medicine (permit NoEP 108/2017) and conducted according to the European Communities Council Directive and the International Association for the Study of Pain guidelines [50]. All efforts were made to reduce the number of animals. All data was presented in accordance with the ARRIVE guidelines [51].

### 4.1. Persistent Immunogenic Hypersensitivity (PIH) in the TMJ

Forty male Sprague–Dawley rats (300–400 g, age 6–8 weeks at the start of experiment, University of Zagreb School of Medicine, Zagreb, Croatia) were housed in plastic cages with a maximum of 3 animals per cage. Animals were kept in a room with a controlled constant temperature (22 ± 0.5 °C) and a 12 h:12 h light:dark cycle (lights on from 07:00 to 19:00. Food and water were available ad libitum.

Animals were induced to PIH in the TMJ according to previous experiments [8,10] (summarized experimental design on Figure 8). On day 0, the rats were immunized systemically with methylated bovine serum albumin (mBSA, 500 μg; Sigma, St. Louis, MO, USA) used as the antigen, diluted in an emulsion containing immunologic response enhancer complete Freund´s adjuvant (CFA, 100 μL; Sigma, St. Louis, MO, USA) containing inactivated *Mycobacterium tuberculosis* in mineral oil, and 100-μL phosphate-buffered saline (PBS) injected subcutaneously (s.c.) into their back (day 0). On days 7 and 14, animals received further booster s.c. injections (aimed at other parts of the back) containing additional 500-μg doses of mBSA, in which CFA was replaced by incomplete Freund´s adjuvant (IFA, 100 μL; Sigma, St. Louis, MO, USA) devoid of inactivated Mycobacterium. After systemic immunization with mBSA and CFA/IFA, to induce a more localized TMJ monoarthritis, the animals further received unilateral (left) intra-TMJ injections of the low-dose antigen (mBSA; 10 μg, i.a.) dissolved in PBS (15 μL) on days 21, 28, and 35.

In comparison to PIH-induced animals, the control animals (noninduced to PIH) were similarly treated with mBSA; however, they were not subjected to systemic immunologic adjuvants CFA and IFA. These rats received a s.c. injection of mBSA (500 μg) diluted only in phosphate-buffered saline (PBS, 100 μL) on days 0, 7, and 14. Then, similar to PIH rats, on days 21, 28, and 35, the animals received a unilateral (left) intra-TMJ injection of sterile mBSA (10 μg; i.a.) dissolved in PBS (15 μL).

### 4.2. Study Design

Animals were randomly allocated into four different experimental groups (*n* = 10). The first group included animals noninduced to PIH that received i.a. injection of saline on day 42. The second group included animals induced to PIH and treated with saline solution into the TMJ on day 42. The third and the fourth groups included animals induced to PIH and treated with either abobotulinumtoxinA or onabotulinumtoxinA into the TMJ on day 42.

AbobotulinumtoxinA (aboBoNT-A; Dysport^®^, Ipsen, Wrexham, UK) or onabotulinumtoxinA (onaBoNT-A; Botox^®^, Allergan Inc., Irvine, CA, USA) was injected i.a. into the ipsilateral TMJ (left side) of anesthetized animals (20 μL, saline-diluted, using a 30-gauge needle) on day 42 of the experiment (Figure 1). The applied doses (14 U/kg for aboBoNT-A, equivalent to 30.1 pg/400 g animal and 7 U/kg for onaBoNT-A, equivalent to 25.2 pg/400 g animal) were chosen according to previously employed doses in our laboratory [8] and a conventional dose conversion ratio between aboBoNT-A and onaBoNT-A [45,46,47]. Sterile saline solution (NaCl 0.9%) was used as a control treatment for noninduced and induced groups.

### 4.3. Spontaneous and Mechanically Evoked Nocifensive Behaviors

All behavioral evaluations were conducted before (day 13; pre-formalin) and after a low dose of intra-TMJ formalin injection (day 14, post-formalin) in alignment with the time points used in our previous experiment (Figure 1) [8]. Animals were allowed to accommodate for 10 min inside the testing cages before each evaluation in a quiet environment with the appropriate lighting (between 9 a.m. and 4 p.m.). For the pre-formalin assessments, animals were placed directly in the testing cage. For the post-formalin assessment, animals were briefly anesthetized with isoflurane (5% induction; 2.5% maintenance), injected i.a. with low-dose formalin (0.5%; 15 μL), and immediately returned to the testing cage to recover (recovery time: 1 to 2 min) and to perform the behavioral evaluations in the following order: (1) behavioral nocifensive responses and rat grimace scale evaluated simultaneously and (2) mechanical sensitivity using von Frey filaments. The experimenter who performed the behavioral assessments was blinded to all treatments.

#### 4.3.1. Behavioral Nocifensive Responses

The spontaneous nocifensive responses were assessed for 45 min, as described by previous studies [8,12]. The assessment of the behavioral nocifensive responses was calculated by the sum of seconds the animal spent rubbing or scratching the orofacial region (assessed live by a stopwatch) plus the total number of head flinches (1 flinch = 1 s).

#### 4.3.2. The Rat Grimace Scale (RGS)

A video camera was placed in the upper part of the testing cage in a position that allowed a complete view of each animal. Rats were continuously recorded during the evaluation of their behavioral nocifensive responses (starting after awakening from isoflurane anesthesia) divided across 3 blocks of 5 min with a 4-min interval between each block (0–5′, 9–14′, and 18–23′). After recording, the videos were analyzed by an experimenter blinded to the treatment. A 0–2 score (0 = not present, 1 = moderate, and 2 = obvious pain) was assigned to each facial parameter (orbital tightening, nose/cheek flattening, ear changes, and whisker changes) observed in each 5-min block, as previously described [52]. A mean of the scores in all blocks was considered the total RGS score.

#### 4.3.3. Mechanical Sensitivity to von Frey Filaments

Mechanical allodynia was assessed immediately after the completion of the assessment of spontaneous behavioral nociceptive responses by using von Frey monofilaments (Stoeling Co., Wood Dale, IL, USA). The assessment of mechanical allodynia lasted for 5–10 min. The test was performed as previously described [29]. The filaments (flat contact area and weights in grams: 0.16, 0.4, 0.6, 1, 1.4, 2, 4, 6, 8, and 10) were applied bilaterally over the TMJ area, starting with the contralateral nontreated TMJ (right TMJ). In each session, each filament was applied three times (starting with 0.16-g filament and continuing in ascending order) until a repeated positive response (escape and/or defensive movement) after the stimulation of the evaluated area was elicited. The maximum weight (10 g) was assigned when no response was observed.

### 4.4. Immunohistochemistry

On day 57, immediately after behavioral assessments, all animals were deeply anesthetized (ketamine 70 mg/kg and xylazine 7 mg/kg, intraperitoneally (i.p.)) and transcardially perfused with saline followed by buffered 4% paraformaldehyde (PFA) in PBS (pH 7.4). The brain was dissected and placed in the PFA solution overnight (18–20 h) at +4 °C. Then, the samples were transferred to 30% sucrose dissolved in PBS for 48h at +4 °C. Then, all samples were stored at −80 °C prior to further use. The caudal part of the medulla oblongata containing the trigeminal nucleus caudalis (TNC; from −5.28 mm to −6.72 interaural, based on Paxinos and Watson [53]) was cut into 35-μm coronal cross-sections using a cryostat. The position of the sections relative to the bregma/interaural line was determined based on visible landmarks such as the obex, position of the central canal, and shape of the white and gray matter, according to the rat brain atlas [53]. For each immunohistochemical analysis, 5 sections per animal were randomly chosen.

#### 4.4.1. Localization of cSNAP-25 in the Brainstem

Sections corresponding to the TNC region of the brainstem were randomly selected, washed in PBS with 0.25% Triton X-100 (PBST), and exposed to 50 μL of 3% peroxide diluted in 200-μL PBS for 1h. Sections were washed again and blocked with 10% normal goat serum (NGS) in PBST. A well-characterized rabbit polyclonal antibody that specifically cleaves the BoNT-A-truncated SNAP-25 (nonaffinity-purified rabbit antiserum (anti-SNAP-25_197_, National Institute for Biological Standards and Control, Potters Bar, UK, a generous gift from Dr. Thea Sesardic) was diluted in 1% NGS (1:4000), and samples were incubated overnight at room temperature. The next day, the sections were washed, and 200 μL of Alexa Fluor 555 secondary antibody (1:400, Molecular Probes, Invitrogen, Carlsbad, CA, USA) was used to enhance the cSNAP-25 signal. Samples were then washed 3 times before mounting into the glass slide and coverslipped with an antifading agent. All samples were visualized with an epifluorescent microscope (Olympus BX-51, Tokyo, Japan). Fiber-shaped staining was searched, and pictures were taken using a digital camera (DP-70, Olympus, Tokyo, Japan).

#### 4.4.2. Quantification of Neuronal and Astrocyte Activation in the Brainstem

The brainstem sections of each animal containing the TNC were randomly selected for immunohistochemical analyses. The sections were placed in free-floating plate wells in PBS with PBST and washed and blocked with 10% NGS in PBST. Sections of TNC were incubated overnight at room temperature in 1% NGS with either anti-c-Fos (1:500, Santa Cruz Biotechnology, Dallas, TX, USA), anti-calcitonin gene-related peptide (CGRP, 1:10,000, Sigma, St Louis, MO, USA), or anti-glial fibrillary acidic protein (anti-GFAP; 1:600, Sigma, St. Louis, MO, USA) antibodies. The next day, the sections were incubated with fluorescent secondary antibody (goat anti-rabbit Alexa Fluor 555; Molecular Probes, Invitrogen, Carlsbad, CA, USA/goat anti-rabbit Alexa Fluor 488; Invitrogen, Carlsbad, CA, USA). Sections were washed with PBS mounted on glass slides, and coverslipped with an antifading agent.

Immunostained sections were visualized with an Olympus BX-51 epifluorescent microscope coupled to a DP-70 digital camera (Olympus, Tokyo, Japan), and brain regions were identified using the rat stereotaxic atlas [53]. All images were processed and quantified by using cellSens Dimension Software (Olympus, Tokyo, Japan). The c-Fos-positive neurons were automatically counted in the ipsilateral and contralateral side of the dorsal horns of each section (TNC). For the acquisition of GFAP, microphotographs of each coronal section containing the TNC was taken ipsilaterally at a 10× objective magnification and 0.5× camera adapter, while for CGRP quantification, we employed 4× objective magnification and a 0.5× camera adapter. A constant exposure was used to acquire individual microphotographs for each subsequent quantification, as well as the manual thresholds used for calculation of the object number and immunoreactivity areas. The immunohistochemical quantification of the area of immunoreactivity, as well as the number of objects within the area of interest, were based on manual pixel thresholding of grey images obtained from separated RGB channels: red or green based on the fluorophore used (pixel threshold range 115–256 for GFAP and c-Fos images taken at 10× objective magnification; range 62–256 for CGRP microphotographs taken at 4× objective magnification).

### 4.5. Statistical Analysis

Quantitative data were presented as the mean ± standard error of the mean (SEM). Data from repeated behavioral measurements were analyzed using the two-way repeated measures analysis of variance (RM ANOVA), followed by Bonferroni’s post hoc test to compare the intertreatment effects. Single time observations were analyzed by one-way ANOVA, followed by Tukey’s post hoc test. The data representing mechanical sensitivity, assessed by Von Frey filaments, were log-transformed prior to statistical analysis, according to the fact that mechanical sensation is perceived on a logarithmic scale (Weber’s law) [54].

All statistical analyses were performed with GraphPad Prism version 5 (GraphPad Software, Inc., San Diego, CA, USA). A *p*-value < 0.05 was considered significant.

## 5. Conclusions

Based on the present data and previous results [8], the analgesic activity of BoNT-A in rats exposed to a PIH in the TMJ seems to be associated with peripheral and central sites of BoNT-A action. The present results suggest that BoNT-A action is associated with a reduction of spontaneous and evoked nociceptive pain measures suggestive of central sensitization (reduction of pain grimacing in the later assessment phase similar to the BoNT-A action on phase II of the formalin test and reduction of bilateral mechanical allodynia). In addition, we found that the antinociceptive effect of BoNT-A is accompanied by reduction of neuronal and astrocytic activation in the TNC, both effects associated with central pain processing. The mentioned actions did not significantly differ with respect to different pharmacological preparations of the toxin (aboBoNT-A vs. onaBoNT-A). In addition, the antinociceptive activity of aboBoNT-A and onaBoNT-A was accompanied by the occurrence of central cSNAP-25, suggestive of the toxin’s direct central action. However, the relative importance of each site of action (peripheral vs. central) for this effect remains to be further investigated in the present pain model.

## Figures and Tables

**Figure 1 toxins-14-00161-f001:**
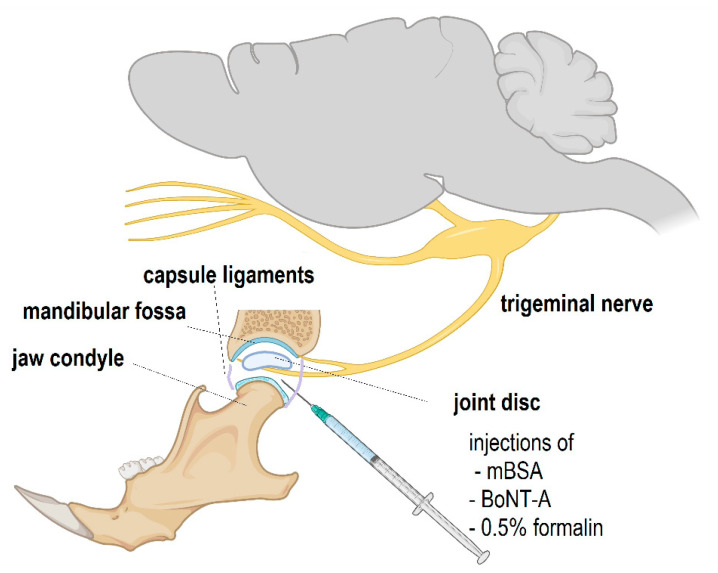
Schematic diagram of the structure of the temporomandibular (TMJ) joint innervated by the trigeminal nerve mandibular branch, as well as the site of injection of experimental substances, such as methylated bovine serum albumin (mBSA), abobotulinumtoxinA, or onabotulinumtoxinA (BoNT-A), and subsequent low-dose formalin challenge. The sizes of structures and their relative positions are not to scale. The illustration was generated by using BioRender© (Biorender.com, assessed on 16 February 2022).

**Figure 2 toxins-14-00161-f002:**
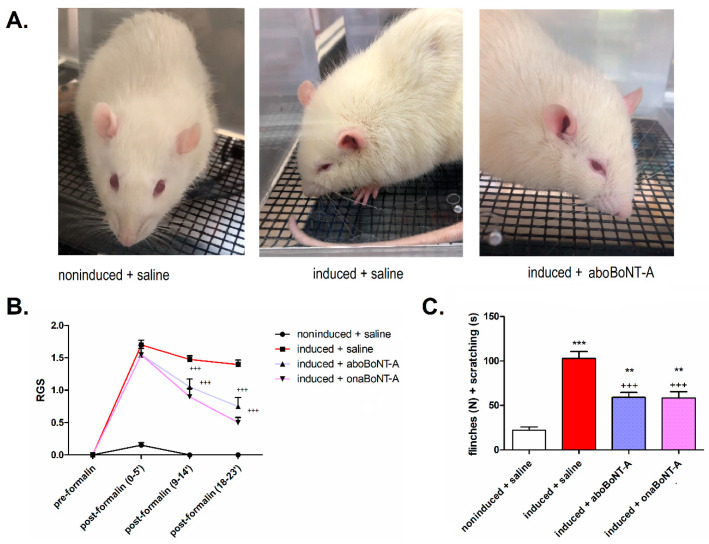
Effect of botulinum toxin type A1 (abobotulinumtoxinA—aboBoNT-A and onabotulinumtoxinA—onaBoNT-A) on spontaneous behavioral responses and the rat grimace scale (RGS) prior to and after low-dose (0.5%) formalin stimulation of the temporomandibular joint (TMJ) of hypernociceptive rats. (**A**) Representative photographs showing rat facial expression in noninduced + saline (noninduced), induced + saline (induced), and induced + abobotulinumtoxinA (aboBoNT-A) animals after formalin stimulation. In the photos, the best visible features of nociceptive grimacing are orbital tightening, as well as characteristic spread-out ear positioning in animals experiencing more intensive pain (induced + saline vs. induced + onaBoNT/A), compared to no observable painful grimacing (noninduced + saline). (**B**) Both BoNT-A pharmaceutical preparations reduced the nociceptive facial grimacing evoked by 0.5% formalin, as assessed by RGS. No face grimacing was observed at the pre-formalin assessments. The individual RGS are calculated as the average of 4 distinctive facial features (orbital tightening, nose/cheek bulging, ear positioning, and whisker positioning) on a scale from 0 to 2. Mean ± SEM; +++ *p* < 0.001 vs. induced + saline (two-way repeated measures ANOVA followed by Bonferroni´s post hoc test). (**C**) Both BoNT-A intra-articular (i.a.) treatments (14 U/kg aboBoNT-A or 7 U/kg onaBoNT-A) reduced the spontaneous nocifensive behaviors (total sum of the number of flinches—N, plus the duration of facial grooming and rubbing—s) evoked by low-dose (0.5%) i.a. formalin in rats exhibiting persistent immunogenic hypersensitivity compared to noninduced animals. N(animals/group) = 10. Mean ± SEM, *** *p* < 0.001 and ** *p* < 0.01 vs. noninduced + saline and +++ *p* < 0.001 vs. induced + saline (F_3,36_ = 49.59, one-way ANOVA followed by Tukey’s post hoc test).

**Figure 3 toxins-14-00161-f003:**
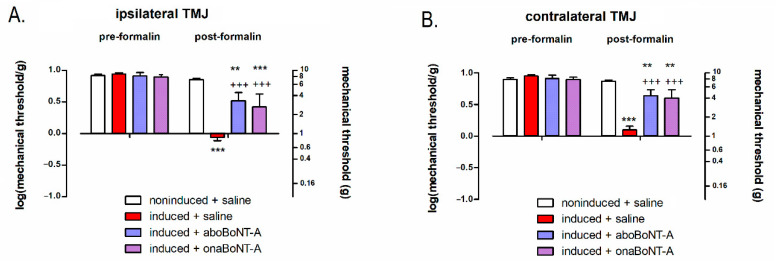
Effects of abobotulinumtoxinA (aboBoNT-A, 14 U/kg) and onabotulinumtoxinA (onaBoNT-A, 7 U/kg) on mechanical allodynia assessed by von Frey filaments in the (**A**) ipsilateral and (**B**) contralateral temporomandibular joint (TMJ) of PIH rats prior to and after i.a. stimulation with 0.5% low-dose formalin. The values were shown as the log-transformed data (log_10_(value in grams)). The values on the right y-axis represent the dose range of filaments used (in grams) on the corresponding logarithmic scale. N(animals/group) = 10; *** *p* < 0.001 and ** *p* < 0.01 vs. noninduced + saline and +++ *p* < 0.001 vs. induced + saline (two-way repeated measures ANOVA followed by Bonferroni’s post hoc test).

**Figure 4 toxins-14-00161-f004:**
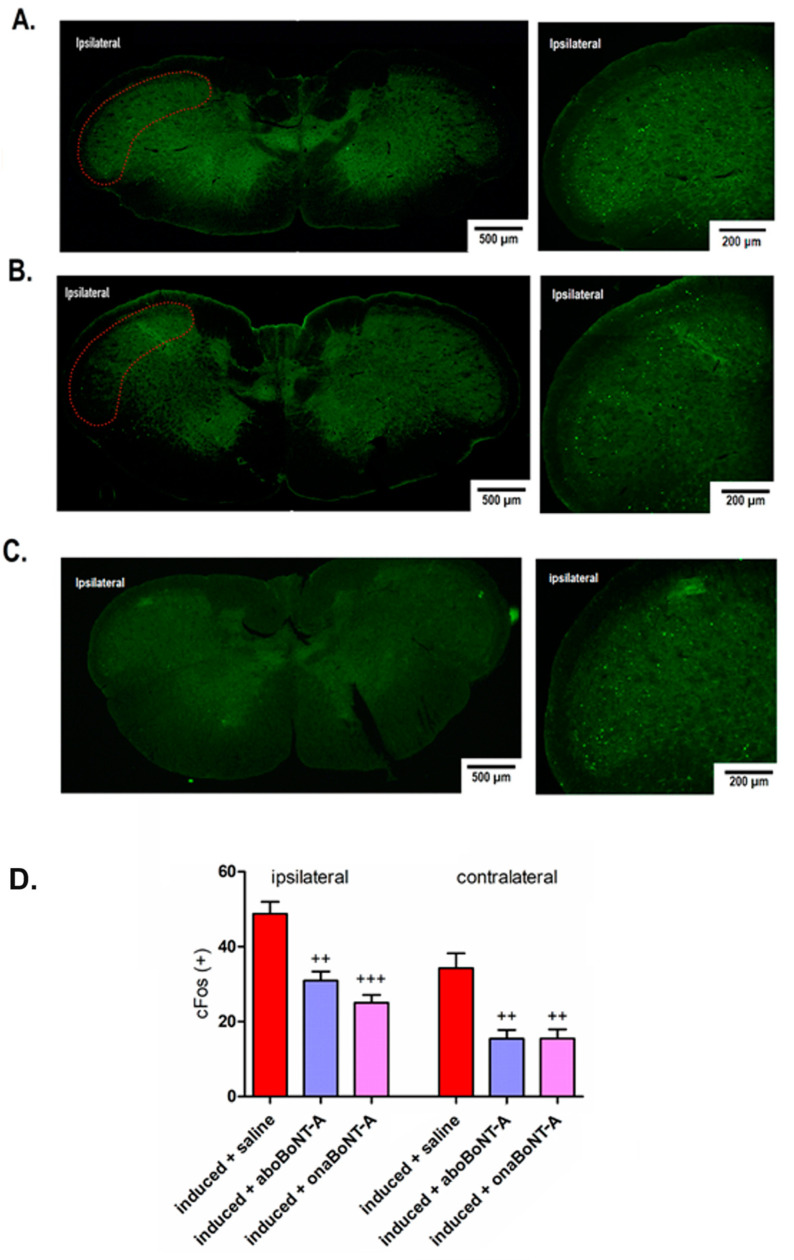
Bilateral effects of intra-articular (i.a.) abobotulinumtoxinA (aboBoNT-A, 14 U/kg) or onabotulinumtoxinA (onaBoNT-A, 7 U/kg) on neuronal activation (c-Fos green punctate immunoreactivity) in the trigeminal nucleus caudalis (TNC) after low-dose (0.5%) i.a. formalin stimulation of rats exposed to the persistent immunogenic hypersensitivity model in the temporomandibular joint. Immunohistochemical staining of c-Fos activation in the TNC of (**A**) induced + saline, (**B**) induced + aboBoNT-A-injected animals, and (**C**) induced + onaBoNT-A animals. The red dotted line indicates the analyzed TNC area. (**D**) Quantification of c-Fos-expressing neuronal profiles (5 sections/animal, N = 5 animals/treatment group). Mean ± SEM; +++ = *p* < 0.001 and ++ = *p* < 0.01 vs. induced + saline (F_5,24_ = 20.1, one-way ANOVA followed by Tukey’s post hoc test).

**Figure 5 toxins-14-00161-f005:**
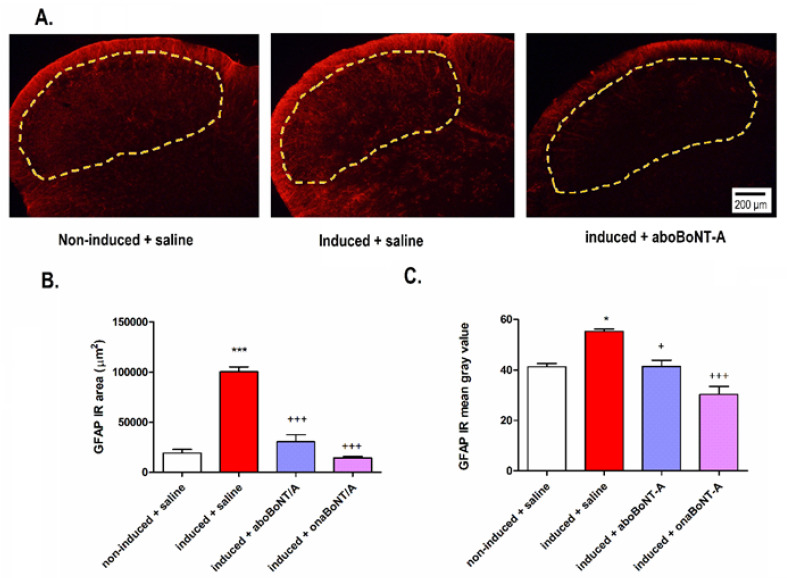
Effects of abobotulinumtoxinA (aboBoNT-A, 15 U/kg) and onabotulinumtoxinA (onaBoNT-A, 7 U/kg) on glial fibrillary acidic protein (GFAP) expression in the trigeminal nucleus caudalis (TNC) of rats exposed to the persistent immunogenic hypersensitivity model in the temporomandibular joint. (**A**) The representative microphotographs of TNC immunostained for GFAP (representative of 5 animals per group). Astrocyte activation was evident as (**B**) an increase in the GFAP-immunoreactive (IR) area and (**C**) increase in the mean intensity (mean gray value) of arthritis-induced animals and prevented by aboBoNT-A and onaBoNT-A. The analysis was performed on 5 randomly selected slices per animal (n = 5 animals/group). The yellow dotted lines indicate the analyzed TNC area. Mean ± SE; *** *p* < 0.001 and * = *p* < 0.05 vs. noninduced + saline and +++ = *p* < 0.001 and + = *p* < 0.05 vs. induced + saline (F_3,16_ (GFAP area) = 73.14; F_3,16_ (GFAP gray value) = 23.21; one-way ANOVA followed by Tukey’s multiple comparisons test).

**Figure 6 toxins-14-00161-f006:**
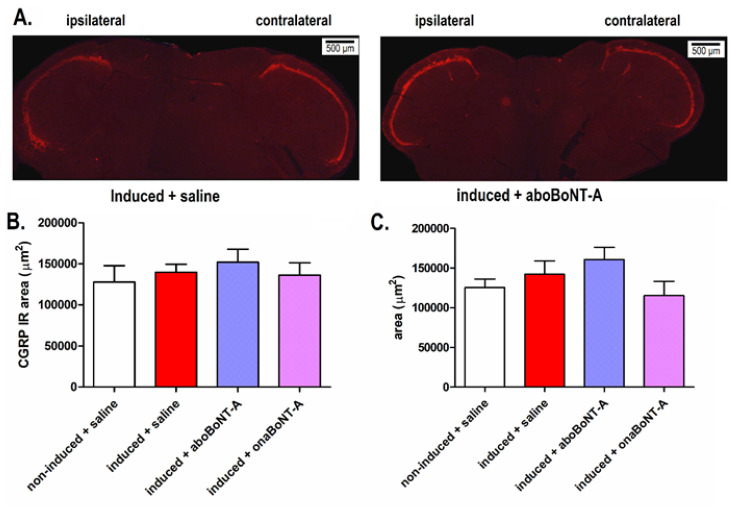
Lack of increase of CGRP immunoreactivity in the trigeminal nucleus caudalis (TNC) of rats exposed to the persistent immunogenic hypersensitivity model in the temporomandibular joint (**A**). Representative images of TNC sections immunostained for CGRP (representative of 5 animals/group). Quantification of the pixel threshold area of CGRP immunoreactivity in the ipsilateral (**B**) and contralateral (**C**) TNC demonstrated lack of changes of CGRP expression. The analysis was performed on 5 randomly selected slices per animal (n (animals) = 5/group). The yellow dotted lines indicate the analyzed TNC area. Mean ± SE; (F_3,16_ (ipsilateral) = 0.3819; F F_3,16_ (contralateral) = 1.267; one-way ANOVA).

**Figure 7 toxins-14-00161-f007:**
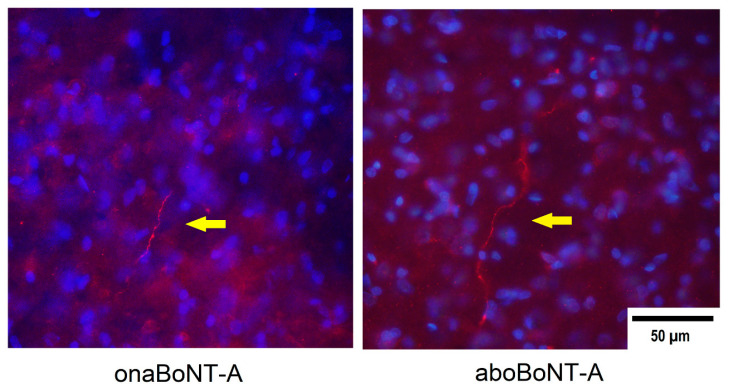
Immunohistochemical staining of cleaved SNAP-25 (cSNAP-25) in the trigeminal nucleus caudalis (TNC) ipsilateral to BoNT-A injections in the temporomandibular joint (TMJ) on day 14 after 7-U/kg onabotulinumtoxinA (onBoNT-A) or 14-U/kg abobotulinumtoxinA (aboBoNT-A) injection. In the epifluorescent microphotographs, yellow arrows indicate cSNAP-25-positive individual fibers (in bright red), while the blue counterstain represents the cell nuclei stained by 4′,6-diamidino-2-phenylindole (DAPI). The sections shown are representative of 10 sections per animal (N = 3/treatment group).

**Figure 8 toxins-14-00161-f008:**
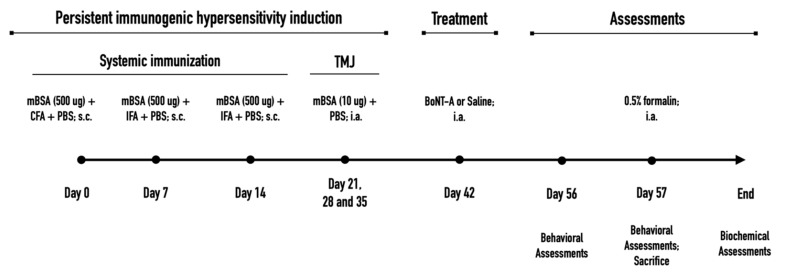
Experimental design indicating the time points of the persistent immunogenic hypersensitivity model, applied treatments, and assessments. mBSA, methylated bovine serum albumin; CFA, complete Freund’s adjuvant; IFA, incomplete Freund’s adjuvant; PBS, phosphate-buffered solution; BoNT-A, botulinum toxin type A1 (abobotulinumtoxinA or onabotulinumtoxinA); TMJ, temporomandibular joint; s.c., subcutaneous; i.a., intra-articular.

## Data Availability

The data presented in this study are available on request from the corresponding author.

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
