# Peer review of "Antinociceptive Actions of Botulinum Toxin A1 on Immunogenic Hypersensitivity in Temporomandibular Joint of Rats"

_toxins, 2022, doi:10.3390/toxins14030161_

Round 1
Reviewer 1 Report
The authors have investigated the effects of two preparations of BoNT-A, abobotulinumtoxinA (aboBoNT-A; Dysport) and onabotulinumtoxinA (onaBoNT-A; Botox), on spontaneous and evoked nociceptive behaviors, as well as on central neuronal and astroglial activation. Both BoNT-A preparations similarly reduced the formalin-induced spontaneous pain-related behaviors and mechanical allodynia of the hypernociceptive rats. Likewise, their effects were associated with the central occurence of cSNAP-25, and reduction of cFos and GFAP up-regulationj in the TNC.
The work has been performed well. The manuscript is extensive and interesting. The overall composition of the manuscript is good. I recommend acceptance of the paper with minor checks.
1) Define abbreviations in the part of the manuscript where they are presented for the first time, since many of them are defined in the Material and Methods section, despite having been mentioned previously. For example:
Line 37: To define NSAIDs
Line 64: To define cSNAP-25
Line 71: To define RGS
Line 121: To define GFAP
Line 123: To define CGRP
Line 80: Effiect. Is correct?
Line 209: To define CFA
Line 193: (9-14 min and 18-23), change for (9-14 and 18-23 min)
2) I recommend that the authors present the results with a brief description of why the experiment was done or what was the purpose of the realization of the experiment.
3) The graphs must be improved. They are seen in low resolution. Example. Figure 1, panels B and C are not defined. Please Verify the resolution of all Figures.
4) Line 182: The authors describe (Figs 1 and 2). Homogenize. All the manuscript are described as Figure.
5) Lines 258-259. Check the form are represented the references in the main document (Scaglione et al. (2016). Following a study from Field et al. (2018)). According to author instructions, these must go in brackets and numbers.
Author Response
Reviewer no. 1.
1) Define abbreviations in the part of the manuscript where they are presented for the first time, since many of them are defined in the Material and Methods section, despite having been mentioned previously. For example:
Line 37: To define NSAIDs
Author response: Corrected
Line 64: To define cSNAP-25
Author response: Corrected
Line 71: To define RGS
Author response: Corrected
Line 121: To define GFAP
Author response: Corrected
Line 123: To define CGRP
Author response: Corrected
Line 80: Effiect. Is correct?
Author response: Corrected to “Effect”
Line 209: To define CFA
Author response: Defined as “complete Freund’s adjuvant”
Line 193: (9-14 min and 18-23), change for (9-14 and 18-23 min)
Author response: Corrected
2) I recommend that the authors present the results with a brief description of why the experiment was done or what was the purpose of the realization of the experiment.
Author response: Accepted
Inserted text:
“Previously in the PIH model, the number of spontaneous motor responses (flinches, scratching) after the formalin stimulation of was shown to be reduced by ona-BoNT-A[8]. Herein, we extended the measurement of spontaneous nociception by examining facial grimacing related to pain (RGS), and examined if the spontaneous nociceptive responses are present both before and after the TMJ stimulation with low-dose formalin 0.5%).
…
“By employing the von Frey filaments, we examined the possible occurence of facial mechanical allodynia over the skin covering the stimulated TMJ area, both before and after the stimulation with 0.5% formalin.
…
We further examined possible BoNT-A effect on neuronal and astrocyte activation associated with nociception in the trigeminal nucleus caudalis (TNC), the first order pain processing sensory nucleus of the cranial area.
…
Previously, we discovered that the BoNT/A antinociceptive action in the craniofacial area is associated with the occurrence of cleaved SNAP-25 (cSNAP-25) in the TNC following axonal transport[28,32]. Thus, we further examined if the antinociceptive actions of aboBoNT/A and onaBoNT/A are similarly associated with the direct toxin activity in the TNC.
3) The graphs must be improved. They are seen in low resolution. Example. Figure 1, panels B and C are not defined. Please Verify the resolution of all Figures.
We believe that, unfortunately, the resolution of our images is greatly reduced by the process of file conversion (from Word to PDF) itself, which cannot be influenced by us. We have therefore now provided the figures as individual TIFF images with the resolution required for publication. The reviewers are kindly asked to check for the individual figure files in TIFF format, or the word version submitted, if possible.
4) Line 182: The authors describe (Figs 1 and 2). Homogenize. All the manuscript are described as Figure.
Author response: Accepted/Corrected
5) Lines 258-259. Check the form are represented the references in the main document (Scaglione et al. (2016). Following a study from Field et al. (2018)). According to author instructions, these must go in brackets and numbers.
Author response: Accepted/Corrected
Reviewer 2 Report
Specific comments:
- Page 1, Line 38: A schematic diagram of the composition of the TMJ compartment should be provided for clarity to illustrate the signaling pathway (Lines 44-53).
- Page 2, Line 45: “since it is induced at a specific location” – how the special location is relevant here? How did they know the location?
- Page 2, Lines 67-69: The difference between “aboBoNT-A”(Dysport) and onaBoNT-A” (BOTOX®) should be given (Gene 2003 Oct 2;315:21-32.doi: 10.1016/s0378-1119(03)00792-3) (doi: 10.1111/j.1468-1331.2011.03560.x.) (doi: 10.1016/j.neuro.2005.01.017).
- Fig 1. “Botulinum neurotoxin type A1 (BoNT-A)” Which one of aboBoNT-A”(Dysport) and onaBoNT-A” (BOTOX®) did they use? Why did they use one type?
- Fig 1: “Representative photographs showing rat facial expression” – What specific spots did they expect the reader to see? How did they quantify it? Fig 1B & C came in low resolution, hard to evaluate. A rough read of both Fig 1B & 1C, it seems that there were different effects between aboBoNT-A”(Dysport) and onaBoNT-A” (BOTOX®) as assessed RGS on Panel Fig1B, but none in panel Fig1C. Any explanation for the differential impacts?
- Page 4, Line 99: “Figure 2. Effects of abobotulinumtoxinA (aboBoNT-A, 14 U/kg) and onabotulinumtoxinA (onaBoNT-A, 7 U/kg)” – Why did they use different dosages? On how many rats? How many times of trials? Time intervals?
- Fig 3B: why did they show only the aboBoNT-A”(Dysport)? What happened to onaBoNT-A (BOTOX®)? What was their reference of the trigeminal nucleus caudalis vs. c-Fos?
- How did they register the exact location of Fig 3 and Fig 4 for comparison of an ipsilateral GFAP immunoreactivity vs. c-Fos (How did they compare the intensity of different fluorescence)? How did they quantify either one? By what reference points did they draw “Yellow-dotted lines indicate the analyzed TNC area?”
- Fig 4, Line 129: “The astrocyte activation was evident as B.) increase in” given the poor resolution of the images; it is hard to see what the authors claimed.
- Fig 5A. Given the poor resolution of the images, it is hard to see what the authors claimed.
- Fig 6: The images are pixelated – Line 152 “White arrow indicates cSNAP-25 positive nerve fiber” – any co-labeling confocal immunofluorescence microscopy data to confirm the nature of this staining? What is the nature of the blue staining? Given both Ipsi and contra staining patterns, how did they tell which red staining is real without referencing benchmarks?
- Lines 206-208: “an occurrence of c-Fos activated neurons in the ipsilateral and contralateral TNC of arthritic animals after 0.5% formalin stimulation (Fig 4).” Without any references (controls), it is hard to claim this point. They cited Reference #40 – [Marinelli, S.; Vacca, V.; Ricordy, R.; Uggenti, C.; Tata, A.M.; Luvisetto, S.; Pavone, F. The Analgesic Effect on Neuropathic Pain of Retrogradely Transported botulinum Neurotoxin A Involves Schwann Cells and Astrocytes. PLoS One 2012, 7, e47977, 532. doi:10.1371/journal.pone.0047977]. They might look at Fig 2: The high resolution of co-labeling magnification confocal images articulate the point. So did this PLoS One article repeatedly demonstrate the point of co-localization (Fig 3, 4, 5, 6), essential to identify if they got neurons or astrocytes.
- Line 151: Figure 6. Immunohistochemical staining. They got Line 310 (page 12) “Figure 6. Experimental design indicating time-points of persistent immunogenic” – Both labeled as Fig 6.
- They wrote 40 rats (4 groups); however, they showed only the data pf 3 groups: Fig 1A, Fig 3,
- Lines 434-439: The conclusion did not fully reflect their data.

Author Response
Responses to Reviewer no. 2.
- Page 1, Line 38: A schematic diagram of the composition of the TMJ compartment should be provided for clarity to illustrate the signaling pathway (Lines 44-53).
Author response: Accepted
We inserted a new figure with schematic representation of the TMJ.
- Page 2, Line 45: “since it is induced at a specific location” – how the special location is relevant here? How did they know the location?
Author response: Comment
We meant that the local intra-articular injections of methylated BSA induced, in turn, the local presentation of the joint hypersensitivity. To specify this, we exchanged the mentioned text with:… “ locally inside the mBSA-stimulated joint”
- Page 2, Lines 67-69: The difference between “aboBoNT-A”(Dysport) and onaBoNT-A” (BOTOX®) should be given (Gene 2003 Oct 2;315:21-32.doi: 10.1016/s0378-1119(03)00792-3) (doi: 10.1111/j.1468-1331.2011.03560.x.) (doi: 10.1016/j.neuro.2005.01.017).
Author response: Accepted/Comment
We pointed out the difference between the two toxins related to their composition and the recommended conversion rates in the discussion.
New text (lines 274-278):
Although both aboBoNT-A and onaBoNT-A are produced from identical Hall A strain synthesizing the A1 toxin subtype, they differ in the composition of complexing proteins (onaBoNT-A is composed of the entire 900 kDa complex, while aboBoNT-A has a variable composition of complexing proteins per neurotoxin molecule), as well as the relative potency of the different products[46].
- Fig 1. “Botulinum neurotoxin type A1 (BoNT-A)” Which one of aboBoNT-A”(Dysport) and onaBoNT-A” (BOTOX®) did they use? Why did they use one type?
Author response: Corrected
We tested both aboBoNT-A and onaBoNT-A, which was indicated in the figure legend.
- Fig 1: “Representative photographs showing rat facial expression” – What specific spots did they expect the reader to see? How did they quantify it? Fig 1B & C came in low resolution, hard to evaluate. A rough read of both Fig 1B & 1C, it seems that there were different effects between aboBoNT-A”(Dysport) and onaBoNT-A” (BOTOX®) as assessed RGS on Panel Fig1B, but none in panel Fig1C. Any explanation for the differential impacts?
Author response: Accept/Comment
We provided a small description in the figure legend for the readers about the facial features showing the signs of pain.
Inserted text: “ In the photos, the best visible features of nociceptive grimacing are orbital tightening, as well as characteristic spread-out ear positioning in animals experiencing more intensive pain (induced + saline vs induced + aboBoNT/A), compared to no observable painful gromacing (non-induced + saline). (B) Both BoNT-A pharmaceutical preparations reduced the nociceptive facial grimacing evoked by 0.5 % formalin, as assessed by RGS. No face grimacing was observed at pre-formalin assessments. The individual RGS is calculated as the average of 4 distinctive facial features (orbital tightening, nose/cheek bulging, ear positioning, whisker positioning) on the scale from 0-2.”
-Although the graph might suggest that onaBoNT-A is slightly more efficient, the statistical analysis between the two groups by a post hoc test did not reveal a significant difference.
- Page 4, Line 99: “Figure 2. Effects of abobotulinumtoxinA (aboBoNT-A, 14 U/kg) and onabotulinumtoxinA (onaBoNT-A, 7 U/kg)” – Why did they use different dosages? On how many rats? How many times of trials? Time intervals?
Author response: Comment
We provide in the manuscript the reasoning behind the choice of different doses (Discussion, lines 291-295):
To compare the effect of aboBoNT-A vs onaBoNT-A, we employed 2:1 dose conversion ratio according to Scaglione et al. [45]. According to study from Field et al., who calculated the picogram content of each international unit (U) (5.38 pg of 150 kDa neurotoxin protein for aboBoNT-A, while 1 U of onaBoNT-A contains 9 pg 150kDa protein) [47], approx. 400 g rats were herein injected with similar 150 kDa BoNT-A amounts (30.1 pg/animal of aboBoNT-A and 25.2 pg/animal of onaBoNT-A).
The applied doses (14 U/kg- for aboBoNT-A, equivalent to 30.1 pg/400 g animal; and 7 U/kg for onaBoNT-A, equivalent to 25.2 pg/animal) were chosen according to previously employed doses in our laboratory [8] and a conventional dose conversion ratio between aboBoNT-A and onaBoNT-A [45-47].
In behavioral analyses we employed 10
- Fig 3B: why did they show only the aboBoNT-A”(Dysport)? What happened to onaBoNT-A (BOTOX®)? What was their reference of the trigeminal nucleus caudalis vs. c-Fos?
Author response: Accept/Comment
In the figure 4C we showed the representative image from onaBoNT-A treated rat, as well.
As stated in the text: The reference points for positioning of the trigeminal nucleus caudalis were determined based on visible landmarks such as obex, position of the central canal, shape of the white and gray matter according to the rat brain atlas (53).
- How did they register the exact location of Fig 3 and Fig 4 for comparison of an ipsilateral GFAP immunoreactivity vs. c-Fos (How did they compare the intensity of different fluorescence)? How did they quantify either one? By what reference points did they draw “Yellow-dotted lines indicate the analyzed TNC area?”
Author response: Accept/Comment
Added in the text: To identify the ipsilateral vs. contralateral side relative to the treated TMJ, during the cryostat cutting a small notch by using the tip of a scalpel was repeatedly made on the surface of the coronally cut medulla oblongata near the ventromedial surface of the section (contralateral to the treatments), which was subsequently identifiable in all coronally cut sections.
As previously mentioned, the outline borders of the TNC are easily identified since the characteristical features of medullary dorsal horn are easily identified on the low magnification (4X).
Regarding the mode of quantification, we describe it in Materials & Methods:
Immunostained sections were visualized with an Olympus BX-51 epifluorescent mi-croscope coupled to DP-70 digital camera (Olympus, Tokyo, Japan), and brain regions were identified using the rat stereotaxic atlas [53]. All images were processed and quanti-fied by using the cellSens Dimension Software (Olympus, Tokyo, Japan). The c-Fos posi-tive neurons were automatically counted in the ipsilateral and contralateral side of the dorsal horns of each section (TNC). For the acquisition of GFAP, microphotographs of each coronal section containing the TNC was taken ipsilaterally at a 10 x objec-tive-magnification and 0.5 x camera adapter, while for CGRP quantification we employed 4 x objective magnification and 0.5 x camera adapter. A constant exposure was used to acquire individual microphotographs for each subsequent quantification, as well as the manual thresholds used for calculation of the object number and immunoreactivity areas. The immunohistochemical quantification of the area of immunoreactivity, as well as the number of objects within the area of interest was based on manual pixel thresholding of grey images obtained from separated RGB channels- red or green based on the fluorofore used (pixel threshold range 115-256 for GFAP and c-Fos images taken at 10x objective magnification; range 62-256 for CGRP microphotographs taken at 4 x objective magnifica-tion).
- Fig 4, Line 129: “The astrocyte activation was evident as B.) increase in” given the poor resolution of the images; it is hard to see what the authors claimed.
- Fig 5A. Given the poor resolution of the images, it is hard to see what the authors claimed.
Author response: Accept/Comment
We believe that (as we stated also in response to the reviewer no. 1) unfortunately, the resolution of our images is greatly reduced by the process of file conversion (from Word to PDF) itself, which cannot be influenced by us. We have therefore now provided the figures as individual TIFF images with the resolution required for publication. The reviewers are kindly asked to check for the individual figure files in TIFF format, or the word version submitted, if possible.
- Fig 6: The images are pixelated – Line 152 “White arrow indicates cSNAP-25 positive nerve fiber” – any co-labeling confocal immunofluorescence microscopy data to confirm the nature of this staining? What is the nature of the blue staining? Given both Ipsi and contra staining patterns, how did they tell which red staining is real without referencing benchmarks?
Author response: Comment
Herein, our referencing benchmarks have been in fact the contralateral TNC area of the same animals or the TNC area of control animals which show no specific staining, and the levels of red staining attributable to normal background fluorescence/ autofluorescence of the stained tissue.
In present experiment we have not performed any colocalization analyses since these issues were resolved in our previous studies. After BoNT/A unilateral injections in the orofacial area in rats, we found that the red staining in fiber-like structures was shown to occur in the ipsilateral TNC of BoNT-A treated animals (Matak et al., 2011; Matak et al., 2014). The confocal analyses on the exact cellular localization of the cleaved SNAP-25 present in the TNC were also performed, suggesting that the cleaved SNAP-25 is present in synaptophysin-positive terminals, however, not in MAP-2-positive dendrites or GFAP-expressing astrocytes (Matak et al., 2014). Moreover, we demonstrated that these terminals are TRPV1-expressing central afferent terminals (by employing both colocalization and pharmacological desensitization of trigeminal ganglion by high dose capsaicin). We obtained similar results regarding the cSNAP-25 neuronal localization also in the ventral horn (Matak et al., 2012; Caleo et al, 2018). In the new manuscript version we provide better resolution images of individual fibers.
- Lines 206-208: “an occurrence of c-Fos activated neurons in the ipsilateral and contralateral TNC of arthritic animals after 0.5% formalin stimulation (Fig 4).” Without any references (controls), it is hard to claim this point. They cited Reference #40 – [Marinelli, S.; Vacca, V.; Ricordy, R.; Uggenti, C.; Tata, A.M.; Luvisetto, S.; Pavone, F. The Analgesic Effect on Neuropathic Pain of Retrogradely Transported botulinum Neurotoxin A Involves Schwann Cells and Astrocytes. PLoS One 2012, 7, e47977, 532. doi:10.1371/journal.pone.0047977]. They might look at Fig 2: The high resolution of co-labeling magnification confocal images articulate the point. So did this PLoS One article repeatedly demonstrate the point of co-localization (Fig 3, 4, 5, 6), essential to identify if they got neurons or astrocytes.
- Author response: Accept/Comment
In present experiment, we acknowledge that we are unsure whether the occurrence of cFos is dependent on arthritis.” Therefore, we deleted the line “of arthritic animals” and “on arthritic animals”.
In our experiment, we did not have absolute controls (all animals were treated i.a. with 0.5% formalin). However, our previous data performed on absolute controls (no formalin treatment) in multiple experiments showed no expression of cFos in TNC or spinal cord dorsal horn, and no cFos / very low contralateral expression in formalin-injected animals (Drinovac et al., 2013; Matak et al., 2014). The focus in present experiment was on the possible activity of onaBoNT-A and aboBoNT-A on neuronal activation in the sensory nociceptive nuclei. If reviewer deems it necessary, we could perform repeated staining by including the non-induced and absolute control animals, however, this would require prolongation of present revision.
Regarding the colocalization with GFAP, we previously found that BoNT-A-cleaved SNAP-25 was present in TRPV1-expressing central afferent terminals, and not in the GFAP-expressing astrocytes (Matak et al., 2014; explained in the response above).
- Line 151: Figure 6. Immunohistochemical staining. They got Line 310 (page 12) “Figure 6. Experimental design indicating time-points of persistent immunogenic” – Both labeled as Fig 6.
Author response: Accept/Corrected
- They wrote 40 rats (4 groups); however, they showed only the data pf 3 groups: Fig 1A, Fig 3,
Author response: Comment
In Figure 1A we left out the picture of aboBoNT-A since the aim was to illustrate the effect of either of the toxin preparation on the rat facial expression. We provide the data quantification for all groups in the graphs Figure 1B and 1 C. Regarding the Fig 3 we previously commented on the reasoning for leaving out the control group from analyses.
- Lines 434-439: The conclusion did not fully reflect their data.
Author response: Accept/Comment
We expanded the conclusion for a better explanation, also in line with suggestion from reviewer no. 3
Based on the present data and previous results [8], the analgesic activity of BoNT-A in rats exposed to a PIH in the TMJ seems to be associated to peripheral and central sites of BoNT-A action. Present results suggest that BoNT-A action is associated with reduction of spontaneous and evoked nociceptive pain measures suggestive of central sensitization (reduction of pain grimacing in later assessment phase similar to the BoNT-A action on phase II of formalin test, reduction of bilateral mechanical allodynia). In addition, we found that the antinociceptive effect is accompanied by reduction of neuronal and astrocytic activation in the TNC, both effects associated with central pain processing. Mentioned actions did not significantly differ in respect to different pharmacological preparations of the toxin (aboBoNT-A vs. onaBoNT-A. In addition, the antinociceptive activity of aboBoNT-A and onaBoNT-A was accompanied by the occurrence of central cSNAP-25, suggestive of toxin’s direct central action. However, the relative importance of each site of action (peripheral vs central) for this effect remains to be further investigated in the present pain model.
Reviewer 3 Report
This paper will study how Botulinum neurotoxin type A1 (BoNT-A) reduces the peripheral peptide and cytokine upregulation in rats with antigen-evoked persistent immunogenic hypersensitivity (PIH) of the temporomandibular joint (TMJ). Nocifensive behaviors and central associated protein 25 (cSNAP-25) presence, c-Fos, GFAP and CGRP expression in the trigeminal nucleus caudalis (TNC) were assessed.
The design of the study was appropriate
Authors concluded that peripherally injected BoNT-A reduces spontaneous and mechanically evoked nocifensive behaviors in rats with PIH. Moreover, results indicate that the pain triggered by formalin on arthritic rats activates contralateral trigeminal nociceptive nuclei, explaining the occurrence of bilateral mechanical allodynia (Figures 3 A and B). BoNT-A preparations, injected into the ipsilateral TMJ, reduced both bilateral neuronal activation and the bilateral allodynia in the TMJ area.
However, its analgesic effects were shown to be mediated by axonal transport and associated with toxin’s enzymatic activity in the CNS. After intra articular. injections aimed at TMJ (Fig 6). cSNAP-25 appeared as fiber-like processes in the ipsilateral TNC, which is the primary pain-processing area for the orofacial nociceptive input. In accordance with previous studies, we observed no cSNAP-25 in the contralateral TNC.
This point is interesting and should be discussed.
The results are interesting and the conclusions which are very concise should be broadened and articulated
Author Response
Responses to Reviewer no. 3
However, its analgesic effects were shown to be mediated by axonal transport and associated with toxin’s enzymatic activity in the CNS. After intraarticular. injections aimed at TMJ (Fig 6). cSNAP-25 appeared as fiber-like processes in the ipsilateral TNC, which is the primary pain-processing area for the orofacial nociceptive input. In accordance with previous studies, we observed no cSNAP-25 in the contralateral TNC.
This point is interesting and should be discussed.
Author response: Accept
Inserted text: Previously, we discovered that inhibition of the axonal transport within the peripheral nerve or in the sensory ganglion prevent the antinociceptive effect of peripherally injected BoNT-A, while the toxin injections in the peripheral nerve or in the trigeminal ganglion at doses lower than the ones effective in the periphery produce the antinociceptive effect [15, 28, 29]. Herein, we confirmed the occurrence of BoNT-A (aboBoNT-A and onaBoNT-A) in the brainstem after i.a. injections aimed at TMJ (Fig 6). cSNAP-25 appeared as fiber-like processes in the ipsilateral TNC, which is the primary pain-processing area for the orofacial nociceptive input [43]. In accordance with previous studies, we observed no cSNAP-25 in the contralateral TNC, in line with our observation that the toxin is axonally transported primarily via sensory neurons ipsilateral to BoNT-A injection site and their central projections [32].
The results are interesting and the conclusions which are very concise should be broadened and articulated
Author response: Accept/Comment
We expanded the conclusion section, also in line with the suggestion from reviewer no. 2.
Based on the present data and previous results [8], the analgesic activity of BoNT-A in rats exposed to a PIH in the TMJ seems to be associated to peripheral and central sites of BoNT-A action. Present results suggest that BoNT-A action is associated with reduction of spontaneous and evoked nociceptive pain measures suggestive of central sensitization (reduction of pain grimacing in later assessment phase similar to the BoNT-A action on phase II of formalin test, reduction of bilateral mechanical allodynia). In addition, we found that the antinociceptive effect is accompanied by reduction of neuronal and astrocytic activation in the TNC, both effects associated with central pain processing. Mentioned actions did not significantly differ in respect to different pharmacological preparations of the toxin (aboBoNT-A vs. onaBoNT-A. In addition, the antinociceptive activity of aboBoNT-A and onaBoNT-A was accompanied by the occurrence of central cSNAP-25, suggestive of toxin’s direct central action. However, the relative importance of each site of action (peripheral vs central) for this effect remains to be further investigated in the present pain model.
Round 2
Reviewer 2 Report
R1 is accepted as the authors addressed my critiques.